# Learning Dynamic 3D Gaussians from Monocular Videos without Camera Poses

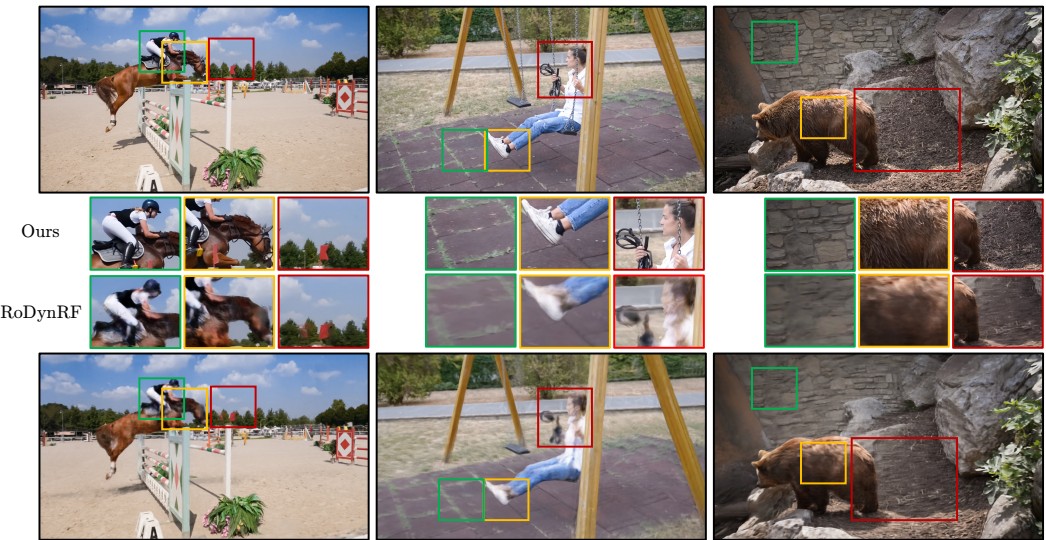

Figure 1: *Mono-DyGS* achieves high-quality reconstruction even under a challenging monocular video without known camera poses. In contrast, RoDynRF (Liu et al., 2023) fails to generate fine details of the given scene.

## Abstract

Dynamic scene reconstruction aims to recover the time-varying geometry and appearance of a dynamic scene. Existing methods, however, heavily rely on the existence of multiple-view captures or the accurate camera poses estimated by Structure from Motion (SfM) algorithms. To relax this constraint, we introduce a method capable of reconstructing generic dynamic scenes, from casually captured monocular videos without known camera poses. Unlike recent works that treat static and dynamic content separately, we propose a unified Hexplane-based Gaussian field to capture the complex effects of scene deformation and camera motion. The Hexplane decomposition enables feasible disentanglement for effective optimization. Combined with an efficient camera pose initialization strategy, our approach significantly improves view synthesis quality and camera pose estimation accuracy over previous methods, while enhancing computational efficiency.

## 1 Introduction

Reconstructing the dynamic scene from a causal video plays a crucial role in understanding and interacting with the real world. Recent studies have made significant strides in modeling complex static 3D scenes (Fu et al., 2024; Kerbl et al., 2023; Chen et al., 2022; Yu et al., 2024b) and dynamic 3D scenes (Liu et al., 2023; Wu et al., 2024; Cao & Johnson, 2023; Gao et al., 2021; Lei et al., 2024; Wang et al., 2024). Most existing methods rely on multiple simultaneous captures with the known camera poses, typically estimated via SfM systems such as COLMAP, as input. However, the multiple-view setting limits their use to causal monocular videos and the SfM systems are not always robust when dealing with dynamic video data due to camera motion blur and the presence of dynamic

objects. Consequently, recovering persistent geometry, radiance, and motion from a monocular video without known camera poses – the most common scenario for in-the-wild data – remains an open and challenging problem.

Recent monocular approaches have demonstrated the ability to operate on casual dynamic videos without known camera poses (Liu et al., 2023; Lei et al., 2024). However, these methods typically rely on disentangling static and dynamic regions using two separate representations. For instance, Liu et al. (2023) employs two distinct TensorRFs (Chen et al., 2022) to model the static and dynamic regions independently where the dynamic TensorRF is trained from scratch using camera poses estimated via the static TensorRF, with no information sharing between the two representations. Since deformation and camera movement occur simultaneously during video captures, modeling them with two separate representations could lead to suboptimal reconstruction results. Furthermore, these methods often suffer from prolonged optimization times due to the random or heuristic initializations for camera poses. For example, RoDynRF requires approximately 20 hours for optimization, while DGMarbles (Stearns et al., 2024) necessitates 5 hours.

Motivated by the above observations, we introduce Mono-DynGS, an algorithm for efficient dynamic scene reconstruction from casual monocular vides. Inspired by the recent success of 3D Gaussians Splatting(3DGS), we represent the dynamic scene as a set of 3D Gaussians for its desired deformable and compositional capability. Instead of using random initialization, we first introduce an efficient camera initialization module by estimating the relative camera pose of every image pair. Concretely, the relative camera pose can be represented by the $\mathbb{SE}(3)$ transformation of a set of 3D Gaussians from the first camera view to the second one. Concurrently, we utilize a deformation field to model the motion of deformable objects from the canonical space to different timesteps. To model complex interactions between scene elements and camera motions, we propose a unified representation shared by both static and dynamic regions. This representation exhibits dual properties: it is unified during rendering for high-quality dynamic reconstruction, yet allows for feasible disentanglement to facilitate effective optimization. In particular, we employ a Hexplane-based encoder for six planes: $\{(x, y), (x, z), (y, z), (x, t), (y, t), (z, t)\}$. During optimization, the first three planes contain information enabling the reconstruction of static backgrounds and camera motion. , the remaining three planes model the underlying deformation and, together with the first three spatial planes, recover the dynamic regions. During inference, all six planes are utilized collectively to obtain high-quality rendering results at different timesteps from arbitrary viewpoints. Furthermore, we incorporate depth and optical flow estimations to regularize the optimization of the proposed Mono-DynGS, thereby enhancing geometric consistency.

We conducted comprehensive experiments across three diverse datasets: DyCheck (Gao et al., 2022), NVIDIA DynamicNeRF (Gao et al., 2021), and MPI Sintel (Butler et al., 2012). Our evaluation focused on two key tasks: dynamic novel-view synthesis and camera pose estimation. We compared our results with previous work, including approaches both with and without known camera poses. Our method consistently outperformed existing techniques in both tasks across all three datasets, demonstrating its robustness and effectiveness in handling a wide range of dynamic scene reconstructions and camera motion estimations.

## 2 RELATED WORK

**Dynamic Novel-view Synthesis**   The modeling of dynamic view synthesis can be generally divided into 2 categories based on how the temporal modeling is handled. The first category generally models dynamic scenes as a 6D-input radiance field which views location, time, and view direction as inputs. For instance, To utilize flow priors, NSFF (Li et al., 2021) adds flow prediction as an auxiliary task and constrains the predicted volume density based on the flow provided. Also, some works are built based on previous static novel synthesis works. For instance, K-Planes (Sara Fridovich-Keil and Giacomo Meanti et al., 2023) and HexPlane (Cao & Johnson, 2023) extend TensorRF (Chen et al., 2022) to dynamic scenes. DyNeRF(Li & Li, 2022) conditions the radiance field on a per-time-instant feature vector which is jointly optimized with the radiance field. However, this category will be incompatible with dealing with dynamic scenes with large deformation due to the lack of explicit representation of object motion.

The second set of models deploys a deformable field to connect the real world with canonical representation. D-NeRF (Pumarola et al., 2021) simply uses a pure MLP combined with sinusoidal

embeddings to represent deformation fields, while TiNeuVox (Fang et al., 2022) imposes multi-resolution grids for interpolation to incorporate deformation in different scales. SWAGS (Shaw et al., 2023) builds on 3DGS (Kerbl et al., 2023) and uses a simple MLP for the deformation field, and CoGS (Yu et al., 2024a) mainly focuses on controllable 3D Gaussian Splatting based on learning dynamic scenes. 4DGS (Wu et al., 2024) incorporates Hexplane as its deformable field. Mosca (Lei et al., 2024) uses DQB interpolation on some key points as its deformable field, while Shape-of-motion (Wang et al., 2024) imposes Fourier transformation to fit the trajectory of dynamic Gaussians. However, most of them fail on casual videos since they require camera poses.

**Camera Pose Estimation under Monocular Video** The second line of related work, mostly consisting of SLAM and SfM systems, aims to reconstruct the scene directly from RGB images by jointly estimating camera parameters and 3D geometries. For example, MonoSLAM (Davison et al., 2007) and ORB-SLAM (Campos et al., 2021) reconstruct point clouds and camera poses with sole images by associating feature correspondences. For SfM systems, Bundler (Snavely, 2008) and COLMAP (Schonberger & Frahm, 2016) provide a method to estimate camera parameters for large image sets. Several methods like (Yen-Chen et al., 2021; Meng et al., 2021; Lin et al., 2021), have developed ideas on estimating camera poses using a NeRF model. (Fu et al., 2024; Fan et al., 2024) shows how to incorporate the rising 3D Gaussian Splatting model with camera pose estimation. For camera pose estimation in dynamic scenes, Lei et al. (2024) uses a cluster-based deformable field to deal with dynamic foregrounds while jointly optimizing camera poses. (Liu et al., 2023) uses a simple MLP as its deformable field and introduces numerous regularizations to enforce geometry consistency. However, most of them use separate representations for static backgrounds and dynamic foregrounds. In contrast, We propose to jointly optimize camera poses and a concise representation of dynamic scenes in an end-to-end manner.

## 3 METHOD

Given a sequence of input images $\{I_i \mid 1 \leq i \leq N\}$ representing a monocular dynamic video, along with the camera intrinsics, our goal is to recover the corresponding camera poses $\{P_i \mid 1 \leq i \leq N\}$ for each frame and generate photo-realistic images for arbitrary novel views and timesteps. To this end, we propose Dy-MonoGS which jointly optimizes a set of 3D Gaussian with continuous deformable files and the corresponding camera poses for all input frames. As illustrated in Fig 2, we begin with the efficient relative pose estimation technique to recover the coarse camera trajectory which we find is a crucial step to facilitate the whole scene optimization (Sec. 3.2). Upon the pose initialization, we further optimize a set of Gaussians with a Hexplane-based encoder by disentangling the static(Sec. 3.3.1) and dynamic aspects(Sec. 3.3.2). Meanwhile, we refine the initial camera poses and improve the geometric consistency of proposed *Mono-DynGS* by leveraging the dense depth and optical flow predictions.

### 3.1 PRELIMINARY: 3D GAUSSIAN SPLATTING

3D Gaussian Splatting (Kerbl et al., 2023) is a differentiable rendering method that performs well in 3D reconstruction tasks. It models a scene through a group of "Gaussians" and "splats" them to the image plane. More specifically, a 3D scene is represented by a gaussian set $\mathcal{G}$, which contains multiple Gaussians

$$g(x) = e^{-\frac{1}{2}\mathcal{X}^T \Sigma^{-1} \mathcal{X}} \tag{1}$$

. For each Gaussian, it's parameterized by its center $\mu \in \mathbb{R}^3$, scale $s \in \mathbb{R}^3$, rotation $q \in \mathbb{SO}(3)$, color $c \in \mathbb{R}^3$ and opacity $\alpha \in [0, 1]$. The covariance $\Sigma$ of the Gaussian can be computed by its scale and rotation:

$$\Sigma = RSS^T R, \tag{2}$$

where $S$ is the diagonal matrix characterized by scaling $s$; $R$ is the rotation matrix correponds to $q$.

When rendering novel views, Gaussians are differentiably splatted to the image plane as follows:

$$\Sigma' = JW\Sigma W^T J^T, \tag{3}$$

where $J$ is the Jacobian of the approximately affine projective transformation and $W$ is the camera view transformation matrix. For a certain pixel, the color $c_i$ and opacity $\alpha_i$ of all the Gaussians are

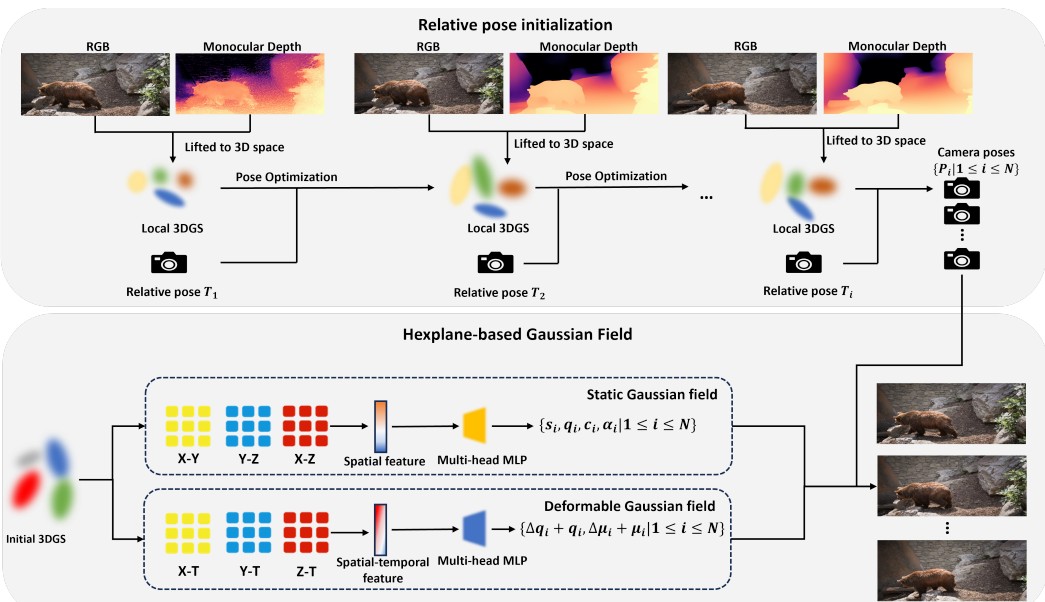

Figure 2: **Overview.** 1) Given a monocular video, we first use the existing dense prediction model to initialize local Gaussians and run relative pose initialization to initialize camera poses(Sec. 3.2); 2) our proposed Hexplane-based Gaussian field model the static geometry and dynamic deformation in a unified representation, through static Gaussian field(Sec. 3.3.1) and triplane deformation(Sec. 3.3.2)

computed using the Gaussian's representation Eq. 1. The blending of N ordered points that overlap the pixel is given by the formula:

$$C = \sum_{i=1}^{N} c_i \alpha_i \prod_{j=1}^{i} (1 - \alpha_i) \tag{4}$$

## 3.2 RELATIVE POSE INITIALIZATION

Some previous works (Fu et al., 2023; Lei et al., 2024) have shown the superiority of 3DGS over implicit representation like NeRFs on recovering camera poses for its explicit representation. Inspired by Fu et al. (2023), we propose to initialize the camera pose trajectory by estimating the relative camera poses between every two adjacent frames. As demonstrated in the top part in Fig. 2, given frame $i - 1$ with image $I_i$, we initialize a set of local Gaussians $G_{i-1}$ by lifting the monocular depth $D_{i-1}$ from a depth prediction model, *i.e.*, Depth Anything (Yang et al., 2024). After the initialization, we first optimize the local 3D Gaussian $G_{i-1}$ based on the current frame $I_{i-1}$

$$G_{i-1}^* = \arg \min_{G_{i-1}} \mathcal{L}(\mathcal{R}(G_{i-1}, \mathbb{I}), I_{i-1}), \tag{5}$$

where $\mathcal{R}$ stands for the differentiable rendering process and $\mathbb{I}$ stands for idenity camera pose. Based on the optimized local Gaussian set $G_{i-1}^*$, we further optimize the relative pose $T_i$ between frame $i$ and frame $i - 1$,

$$T_i^* = \arg \min_{T_i} \mathcal{L}(\mathcal{R}(G_{i-1}^*, T_i), I_i) \tag{6}$$

These two optimizations are only conducted on the static part of images $I_i$ and $I_{i+1}$. By employing the local 3DGS on every pair of images, we can infer the absolute pose based on the first frame as follows,

$$P_i = T_i^* \circ T_{i-1}^* \cdots \circ T_1^* \tag{7}$$

where $\circ$ represents the dot product operation between two camera pose matrices. The optimization of local 3DGS along the whole video is quite efficient, *i.e.*, 80 frames in about 70 minutes. Although these poses could be noisy, we found the coarse trajectory of camera poses performs as a good initialization to accelerate the dynamic scene reconstruction. These initial camera poses are also refined during the following optimization.

### 3.3 HEXPLANE-BASED GAUSSIAN FIELD

Given the initial camera poses, we propose a Hexplane-based Gaussian field to model both the static geometry and dynamic deformation through a unified representation. Recognizing that nearby 3D Gaussians typically share similar spatial and temporal information, we introduce an efficient spatial-temporal encoder that utilizes a 4D Hexplane to decompose the 4D neural voxel into various multi-resolution planes. Specifically, the spatial-temporal structure encoder comprises six plane modules: $\{R_{xy}, R_{xz}, R_{yz}, R_{xt}, R_{yt}, R_{zt}\}$. All 3D Gaussians can be represented by plane features derived from these modules. These six planes can be naturally decomposed into two distinct sets: i) three spatial planes $\{R_{xy}, R_{xz}, R_{yz}\}$; and ii) three temporal planes $\{R_{xt}, R_{yt}, R_{zt}\}$. This decomposition allows us to construct the static Gaussian field using the spatial planes and the deformable Gaussian field using the temporal planes.

#### 3.3.1 STATIC GAUSSIAN FIELD

The proposed static model contains 2 components: gaussian centers $\{\mu_i | 1 \le i \le N_s\}$ and spatial feature fields $\mathcal{F}_s$ composed by three subplane modules $\{R_{xy}, R_{xz}, R_{yz}\}$ and a multi-head Gaussian decoder $\phi$. Each subplane is defined by $R_{ij} \in \mathbb{R}^{N_i \times N_j \times h_s}$, where $h_s$ is the static spatial feature dimension and $N_i, N_j$ denotes the resolution the voxel plane. Given a Gaussian centered at $\mu$, we first perform the bilinear interpolation with the projected 2D coordinates $[p_i, p_j]$ on plane $ij$,

$$f_s^{(ij)} = \text{Interp}\left(R_{ij}, p_i, p_j\right). \tag{8}$$

Then, the final feature is the concatenation of query features on different planes,

$$f_s = \bigcup_{i,j} f_s^{(ij)}, \quad (i,j) \in \{(x,y), (x,z), (y,z)\} \tag{9}$$

where $\bigcup$ represents the concatenation operation. Finally, we compute all Gaussian attributes, *i.e.*, scaling $s$, sphere harmonics $c$, rotation $q$ and opacity $\alpha$, by the multi-head Gaussian decoder $\{\phi_s, \phi_c, \phi_q, \phi_\alpha\}$. Then, the set of Gaussians can be expressed as follows,

$$\mathcal{G} = \{\mu, \phi_s(f_s), \phi_q(f_s), \phi_\alpha(f_s), \phi_c(f_s)\} \tag{10}$$

where we omit the 3D Gaussian index for simplicity.

We jointly optimize the static Gaussian field and the initial 6D camera poses $\{P_i | 1 \le i \le N_t\}$ across different frames, where $N_t$ is the number of input frames. The main supervision signal comes from the photometric loss between the rendered image $\hat{I}_t = \mathcal{R}(\mathcal{G}_s, T_t)$ and the input image $I_t$ at time $t$. We omit time $t$ in the following objective functions for simplicity. To ensure the static field only represents static contents, we directly compute the photometric loss in static regions with the pre-computed mask $M_s$ as follows,

$$\mathcal{L}_{\text{pho}}^s(I, \hat{I}) = M_s \odot \left((1-\gamma)||\hat{I}^s - I^s||_2^2 + \gamma \text{DSSIM}(\hat{I}^s, I^s)\right) \tag{11}$$

where $\odot$ is the element-wise production, DSSIM is the structural dissimilarity loss and we set the factor $\gamma = 0.2$. To address the ill-posed problem inherent in monocular videos, we introduce auxiliary losses to regularize the training process by leveraging estimations of monocular depth and optical flow. Given the forward optical flow $F_i$ and backward flow $B_i$ from frame $i$ to frame $i+1$ and from frame $i$ to frame $i-1$, the reprojection loss is calculated as,

$$\begin{aligned} \mathcal{L}_{\text{reproj}}^s = & \|\pi_{\mathbf{K}}(T_{i+1}T_i^{-1}\pi_{\mathbf{K}}^{-1}(p_i, D_i[p_i])) - (F_i - p_i)\| + \\ & \|\pi_{\mathbf{K}}(T_{i-1}T_i^{-1}\pi_{\mathbf{K}}^{-1}(p_i, D_i[p_i])) - (B_i - p_i)\|, \end{aligned} \tag{12}$$

where $p_i$ denotes the pixel coordinates of the static regions in frame $i$ (we omit the $s$ subscript for simplicity) and $\pi_{\mathbf{K}}$ represents the projection function from 3D space onto the pixel plane using camera intrinsics $K$. To account for potential errors in monocular depth estimation, particularly scale misalignment across different frames, we jointly optimize a correction to depth $D_i$. This correction comprises per-frame global scaling factors and per-pixel adjustments, implemented through a depth alignment loss as follows

$$\mathcal{L}_z^s = D_{\text{inv}}\left(\left[W_{i+1}W_i^{-1}\pi_{\mathbf{K}}(p_i, D_i)\right]_z, D_{i+1}\right), \tag{13}$$

where $[\cdot]_z$ extracts the $z$ coordinate, and $D_{\text{inv}}(x, y) = |\frac{x}{y} - 1| + |\frac{y}{x} - 1|$ is the scale-invariant loss. Consequently, the final loss for the static part is

$$\mathcal{L}^s = \lambda_{\text{pho}}\mathcal{L}_{\text{pho}}^s + \lambda_{\text{reproj}}\mathcal{L}_{\text{reproj}}^s + \lambda_z\mathcal{L}_z^s \tag{14}$$

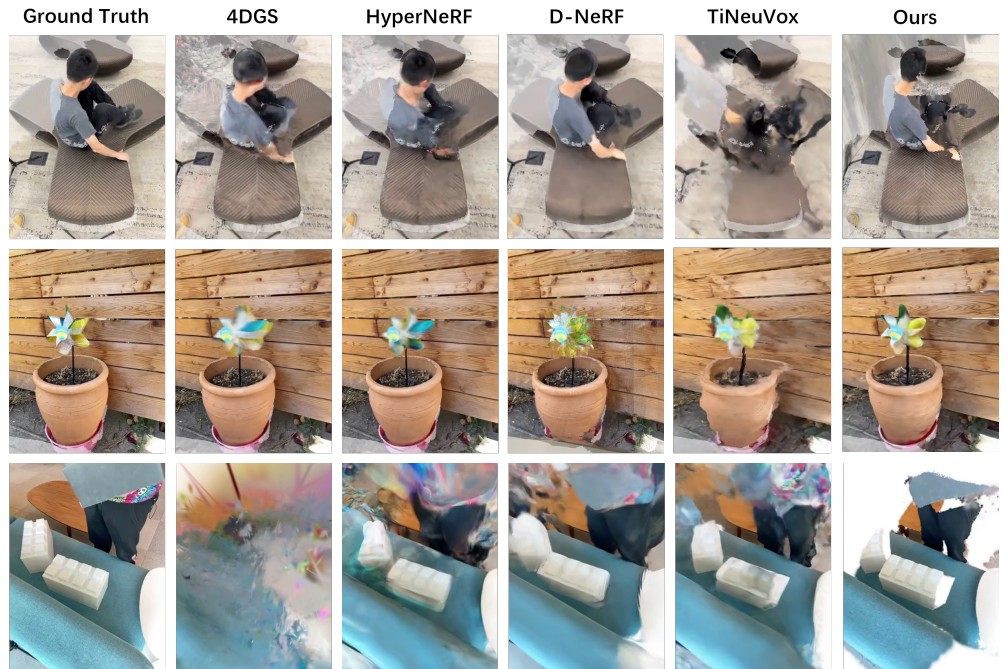

Figure 3: **Novel view synthesis results on iPhone Reality Check dataset.** Note that our model's results are quite sharper than others.

### 3.3.2 DEFORMABLE GAUSSIAN FIELD

We model the deformation of foreground objects by Deformable Gaussian Field composed by three spatial-temporal subplane modules $\{R_{xt}, R_{yt}, R_{zt}\}$ and a multi-head deformation decoder $\xi$. Similar to static Gaussian fields, each subplane has the shape of $N_i \times N_t \times h_d$, where $h_d$ stands for the dimension of spatial-temporal features. Given a 3D Gaussian located at $\mu$, we follow the same procedure to obtain the spatial-temporal feature from three spatial-temporal planes,

$$f_d = \bigcup_{(i,j)} \text{Interp}\,(R_{ij}, p_i, p_j), \quad (i,j) \in \{(x,t),(y,t),(z,t)\} \tag{15}$$

The multi-head deformation decoder is employed to compute the deformation of Gaussian's positions $\Delta\mu = \xi_x(f_d)$ and the Gaussian's rotation $\Delta q = \xi_q(f_d)$. Then, the deformed Gaussian $\mathcal{G}'$ can be expressed as $\mathcal{G}' = \{\mu + \Delta\mu, q + \Delta q, s, \alpha, c\}$, where $q, s, \alpha, c$ are obtained via Eq. 10.

To supervise the deformable Gaussian field, we first employ the same photometric loss $\mathcal{L}^d_{\text{pho}}$ as Eq. 11 while using the foreground dynamic mask $M_d = 1 - M_s$. To enforce the geometric consistency of the deformable Gaussian field, we introduce two regularization terms that focus on the smoothness of rotations and centers of the deformed Gaussians. To maintain the rotation smoothness, we employ the As-Rigid-As-Possible (ARAP) principle. Specifically, we utilize an ARAP loss to generate ground truth rotations corresponding to each 2D tracking trajectory. Note that we utilize Co-tracker (Karaev et al., 2023) here to capture long-range correspondence. We then encourage the deformed Gaussian rotations to align with these ground truth rotations through a rotation smoothness loss. Furthermore, we introduce a spatial smoothness loss to address the temporal consistency of Gaussian centers. This loss encourages the Gaussian centers at different timesteps to remain close to their corresponding 2D tracking trajectories. More details are discussed in A.

## 4 EXPERIMENTAL RESULTS

**Implementation Details.** We use isl-MiDaS (Ranftl et al., 2022) as the backbone to compute monocular depth for each frame. Also, we use CoTracker (Karaev et al., 2023) to compute the long-term pixel trajectory, which facilitates the optimization of camera poses. RAFT (Teed & Deng, 2020) is applied to produce forward and backward optical flows, which is used in fine optimization

|  | apple | | | block | | | paper-windmill | | | space-out | | |
|---|---|---|---|---|---|---|---|---|---|---|---|---|
|  | mPSNR | mSSIM | mLPIPS | mPSNR | mSSIM | mLPIPS | mPSNR | mSSIM | mLPIPS | mPSNR | mSSIM | mLPIPS |
| D-NeRF | 17.43 | 0.728 | 0.508 | 17.52 | 0.669 | 0.346 | 17.55 | 0.367 | 0.258 | 17.71 | 0.591 | 0.377 |
| NSFF | 16.47 | 0.754 | **0.414** | 14.71 | 0.606 | 0.438 | 14.94 | 0.272 | 0.348 | 17.65 | 0.636 | 0.341 |
| 4DGS | 14.44 | 0.698 | 0.716 | 12.30 | 0.498 | 0.706 | 12.77 | 0.251 | 0.697 | 14.46 | 0.479 | 0.790 |
| Shape-of-motion | 16.86 | 0.715 | 0.459 | 16.21 | 0.603 | 0.341 | 16.35 | 0.289 | 0.413 | 16.27 | 0.552 | 0.406 |
| HyperNeRF | 17.64 | **0.743** | 0.478 | 17.54 | 0.670 | 0.331 | 17.38 | 0.382 | 0.209 | 17.93 | 0.605 | 0.320 |
| DynPoint | 17.78 | 0.743 | - | 17.67 | 0.667 | - | 17.32 | 0.366 | - | 17.78 | 0.603 | - |
| PGDVS | 16.66 | 0.721 | 0.411 | 16.38 | 0.601 | 0.293 | 17.19 | 0.386 | 0.277 | 16.49 | 0.592 | 0.326 |
| DyBluRF | 18.00 | 0.737 | 0.488 | 17.47 | 0.665 | 0.349 | 18.19 | 0.405 | 0.301 | 18.83 | 0.643 | 0.326 |
| CTNeRF | **19.53** | 0.691 | - | 19.74 | 0.626 | - | 17.66 | 0.346 | - | 18.11 | 0.601 | - |
| DGSMarbles | 16.50 | 0.703 | 0.499 | 16.11 | 0.599 | 0.363 | 16.19 | 0.302 | 0.454 | 15.97 | 0.513 | 0.437 |
| RoDynRF | 18.73 | 0.722 | 0.552 | 18.73 | 0.634 | 0.513 | 16.71 | 0.321 | 0.482 | 18.56 | 0.594 | 0.413 |
| Mosca | 13.38 | 0.661 | 0.616 | 18.43 | 0.684 | 0.221 | **19.79** | 0.563 | 0.165 | 21.42 | 0.718 | **0.159** |
| Ours | 15.29 | 0.679 | 0.592 | **20.02** | **0.692** | **0.201** | 19.53 | **0.571** | 0.152 | **21.89** | **0.726** | 0.178 |

|  | spin | | | teddy | | | wheel | | | Mean across all scenes | | |
|---|---|---|---|---|---|---|---|---|---|---|---|---|
|  | mPSNR | mSSIM | mLPIPS | mPSNR | mSSIM | mLPIPS | mPSNR | mSSIM | mLPIPS | mPSNR | mSSIM | mLPIPS |
| D-NeRF | 19.16 | 0.567 | 0.443 | 13.71 | 0.570 | 0.429 | 15.65 | 0.548 | 0.292 | 16.96 | 0.577 | 0.379 |
| NSFF | 17.26 | 0.540 | 0.371 | 12.59 | 0.537 | 0.527 | 14.59 | 0.511 | 0.331 | 15.46 | 0.551 | 0.396 |
| 4DGS | 14.93 | 0.417 | 0.640 | 11.86 | 0.458 | 0.729 | 10.99 | 0.304 | 0.803 | 13.11 | 0.443 | 0.726 |
| Shape-of-motion | 17.83 | 0.492 | 0.501 | 13.97 | 0.584 | 0.438 | 15.01 | 0.602 | 0.352 | 15.92 | 0.548 | 0.416 |
| HyperNeRF | 19.20 | 0.561 | 0.325 | 13.97 | 0.568 | 0.350 | 13.99 | 0.455 | 0.310 | 16.81 | 0.569 | 0.332 |
| DynPoint | 19.04 | 0.564 | - | 13.95 | 0.551 | - | 14.72 | 0.515 | - | 16.89 | 0.573 | - |
| PGDVS | 18.49 | 0.590 | 0.247 | 13.29 | 0.516 | 0.399 | 12.68 | 0.429 | 0.429 | 15.88 | 0.548 | 0.340 |
| DyBluRF | 18.20 | 0.541 | 0.400 | 14.61 | 0.572 | 0.425 | **16.26** | **0.575** | 0.325 | 17.37 | 0.591 | 0.373 |
| CTNeRF | 19.79 | 0.516 | - | 14.51 | 0.509 | - | 14.48 | 0.430 | - | 17.69 | 0.531 | - |
| DGMarbles | 17.51 | 0.537 | 0.424 | 13.68 | 0.573 | 0.443 | 14.58 | 0.569 | 0.389 | 15.79 | 0.542 | 0.428 |
| RoDynRF | 17.41 | 0.484 | 0.570 | 14.33 | 0.536 | 0.613 | 15.20 | 0.449 | 0.478 | 17.10 | 0.534 | **0.517** |
| Mosca | 20.20 | 0.650 | 0.188 | 14.40 | 0.573 | 0.314 | 13.04 | 0.399 | **0.314** | 17.24 | 0.607 | 0.283 |
| Ours | **20.82** | **0.661** | 0.186 | **14.98** | **0.585** | 0.307 | 15.32 | 0.488 | 0.513 | **18.26** | **0.629** | 0.304 |

Table 1: **Novel view synthesis results on iPhone reality check dataset**. Each baseline method is trained with its public code under the original settings and evaluated with the given testing poses. The best results are highlighted in bold. According to whether camera poses are necessary during training, we separate the baselines into 2 blocks: the first block contains baselines that require camera poses as input; the second block contains COLMAP-free methods.

of camera poses in the global optimization stage. During inference, we follow the protocol in previous COLMAP-free novel-view synthesis work (Fu et al., 2024), which takes 1 out of 8 frames for inference, while the left 7 frames are used as training data. When testing, we optimize for testing poses that maximize PSNR on testing images, while keeping the Hexplane-based Gaussian field unchanged. We implement our COLMAP-free renderer based on the native 3DGS renderer (Kerbl et al., 2023), which passes gradient to camera pose parameters for pose optimization.

### 4.1 Evaluation on Dynamic View Synthesis

**Results on DyCheck Dataset**   Currently, the most challenging and widely used dataset for monocular reconstruction is the DyCheck dataset (Gao et al., 2022), which is generated by multiple dynamic scenes recorded through a hand-held iPhone device. Also, this dataset provides us with the corresponding camera poses when capturing the videos as well as two static cameras of large baselines for testing. For fairness, we replace our depth predictor based on isl-MiDaS (Ranftl et al., 2022) with the given lidar depth from the dataset. Since the Dycheck dataset provides us with given testing and training views, we choose to apply them for our inference. Most previous 3DGS-based models rely heavily on multi-view stereo cues(present in unnatural fast-moving camera motions) to reconstruct the scene, most of them failed in the DyCheck dataset due to the large deviation of testing camera poses from training camera trajectories. Our model outperforms all existing works in DyCheck scenes as shown in the quantitative results in Tab 1 and the qualitative results in Fig 3 The improvement can be attributed to 2 factors: firstly, our model adopts a 2-stage optimization, which first optimizes in a local Gaussian manner to produce relative poses between frames; and the relative poses are used for the initialization of camera poses, which enables better optimization over the global camera trajectory and facilitates the aggregation of observations over different frames; secondly, our model uses a Triplane to replace the redundant static Gaussian set, which reduces the possibility of overfitting, and since the optimization of Triplane field is much slower than native gaussian attributes, the optimization is done over all frames, which increase the integrity over different timesteps.

**Results on Nvidia Dataset**   We also evaluate our model on the Nvidia dataset, following the inference protocol in RoDynRF (Liu et al., 2023). As shown in Tab 2 and Fig 4, our model reaches highly competitive results on the Nvidia DynamicNeRF dataset. Our improvement is relatively smaller compared to that in the Dycheck dataset due to easier inference settings on the Nvidia dataset: Since the testing and training poses are generated from a single trajectory, the inference is quite

Figure 4: **Novel view synthesis results on Nvidia DynamicNeRF dataset.**

| | Jumping | | | Skating | | | Truck | | | Umbrella | | |
|---|---|---|---|---|---|---|---|---|---|---|---|---|
| | mPSNR | mSSIM | mLPIPS | mPSNR | mSSIM | mLPIPS | mPSNR | mSSIM | mLPIPS | mPSNR | mSSIM | mLPIPS |
| D-NeRF | 17.43 | 0.728 | 0.508 | 17.52 | 0.669 | 0.346 | 17.55 | 0.367 | 0.258 | 17.71 | 0.591 | 0.377 |
| NSFF | 16.47 | 0.754 | 0.414 | 14.71 | 0.606 | 0.438 | 14.94 | 0.272 | 0.348 | 17.65 | 0.636 | 0.341 |
| 4DGS | 17.32 | 0.736 | 0.326 | 19.41 | 0.619 | 0.218 | 21.25 | 0.701 | 0.172 | 19.00 | 0.652 | 0.346 |
| HyperNeRF | 17.64 | 0.743 | 0.478 | 17.54 | 0.670 | 0.331 | 17.38 | 0.382 | 0.209 | 17.93 | 0.605 | 0.320 |
| DynPoint | 17.78 | 0.743 | - | 17.67 | 0.667 | - | 17.32 | 0.366 | - | 17.78 | 0.603 | - |
| PGDVS | 16.66 | 0.721 | 0.411 | 16.38 | 0.601 | 0.293 | 17.19 | 0.386 | 0.277 | 16.49 | 0.592 | 0.326 |
| DyBluRF | 18.00 | 0.737 | 0.488 | 17.47 | 0.665 | 0.349 | 18.19 | 0.405 | 0.301 | 18.83 | 0.643 | 0.326 |
| CTNeRF | 19.53 | 0.691 | - | 19.74 | 0.626 | - | 17.66 | 0.346 | - | 18.11 | 0.601 | - |
| DGMarbles | 19.61 | 0.703 | 0.180 | **24.24** | **0.759** | **0.091** | 27.18 | **0.781** | **0.060** | 23.76 | 0.752 | **0.123** |
| RoDynRF | 18.73 | 0.722 | 0.552 | 18.73 | 0.634 | 0.513 | 16.71 | 0.321 | 0.482 | 18.56 | 0.594 | 0.413 |
| Mosca | 13.38 | 0.661 | 0.616 | 18.43 | 0.684 | 0.221 | 19.79 | 0.563 | 0.165 | 21.42 | 0.718 | 0.159 |
| Ours | **21.01** | **0.752** | **0.109** | 23.65 | 0.732 | 0.108 | **27.52** | 0.769 | 0.106 | **24.58** | **0.741** | 0.136 |
| | Balloon1 | | | Balloon2 | | | Playground | | | Mean across all scenes | | |
| | mPSNR | mSSIM | mLPIPS | mPSNR | mSSIM | mLPIPS | mPSNR | mSSIM | mLPIPS | mPSNR | mSSIM | mLPIPS |
| D-NeRF | 19.16 | 0.567 | 0.443 | 13.71 | 0.570 | 0.429 | 15.65 | 0.548 | 0.292 | 16.96 | 0.577 | 0.379 |
| NSFF | 17.26 | 0.540 | 0.371 | 12.59 | 0.537 | 0.527 | 14.59 | 0.511 | 0.331 | 15.46 | 0.551 | 0.396 |
| 4DGS | 14.11 | 0.309 | 0.404 | 18.56 | 0.607 | 0.239 | 13.51 | 0.457 | 0.341 | 17.59 | 0.583 | 0.292 |
| HyperNeRF | 19.20 | 0.561 | 0.325 | 13.97 | 0.568 | 0.350 | 13.99 | 0.455 | 0.310 | 16.81 | 0.569 | 0.332 |
| DynPoint | 19.04 | 0.564 | - | 13.95 | 0.551 | - | 14.72 | 0.515 | - | 16.89 | 0.573 | - |
| PGDVS | 18.49 | 0.590 | 0.247 | 13.29 | 0.516 | 0.399 | 12.68 | 0.429 | 0.429 | 15.88 | 0.548 | 0.340 |
| DyBluRF | 18.20 | 0.541 | 0.400 | 14.61 | 0.572 | 0.425 | **16.26** | 0.575 | 0.325 | 17.37 | 0.591 | 0.373 |
| CTNeRF | 19.79 | 0.516 | - | 14.51 | 0.509 | - | 14.48 | 0.430 | - | 17.69 | 0.531 | - |
| DGMarbles | 23.65 | **0.698** | 0.072 | 21.60 | 0.791 | 0.142 | **27.18** | **0.804** | **0.060** | 22.32 | **0.756** | 0.129 |
| RoDynRF | 17.41 | 0.484 | 0.570 | 14.33 | 0.536 | 0.613 | 15.20 | 0.449 | 0.478 | 17.10 | 0.534 | 0.517 |
| Mosca | 20.20 | 0.650 | 0.188 | 14.40 | 0.573 | 0.314 | 13.04 | 0.399 | 0.314 | 17.24 | 0.607 | 0.283 |
| Ours | **23.78** | 0.681 | **0.059** | **24.12** | **0.852** | **0.079** | 26.98 | 0.759 | 0.117 | **24.52** | 0.755 | **0.101** |

Table 2: **Novel view synthesis results on Nvidia DynamicNeRF dataset**. Each baseline method is trained with its public code under the original settings and evaluated with the same evaluation protocol. The best results are highlighted in bold. According to whether camera poses are necessary during training, we separate the baselines into 2 blocks: the first block contains baselines that require camera poses as input; the second block contains COLMAP-free methods.

coherent, thus easier than that in the Dycheck dataset. Also, all forward-facing setup reduces the necessity of strong reconstruction on occluded areas.

**Results on Davis Dataset**    We also verify the effectiveness of *Dy-MonoGS* on in-the-wild videos(on DAVIS dataset) in Fig. 5

## 4.2 EVALUATION ON CAMERA POSES ESTIMATION

We conduct camera pose estimation experiments on the MPI Sintel dataset. The results are shown in Table 3. Our model outperforms both previous NeRF-based models, like robust-dynrf (Liu et al., 2023), BARF (Lin et al., 2021), and traditional SfM methods like (Teed & Deng, 2021; Schonberger & Frahm, 2016). The improvement over traditional SLAM methods can be attributed to the global optimization of our model over the entire video instead of local registration over certain frames, which is adopted by SLAM-based methods. Also, the relative camera pose initialization plays an important role in the optimization of camera poses as our model performs better than traditional NeRF-based methods which train all camera poses all at once from scratch.

## 4.3 ABLATION STUDY

To verify the effectiveness of our design, we ablate our full framework. We report the average PSNR, LPIPS, SSIM on the DyCheck dataset and the average ATE and RPE on the MPI Sintel dataset in Tab 4.

We first verify the necessity of relative pose initialization. We can see from the 1st and 2nd rows of Tab 4 that through relative pose initialization, our model's performance on both the dynamic

| Train | Horsejump-high | Swing | Rhino | Bear |
|-------|----------------|-------|-------|------|

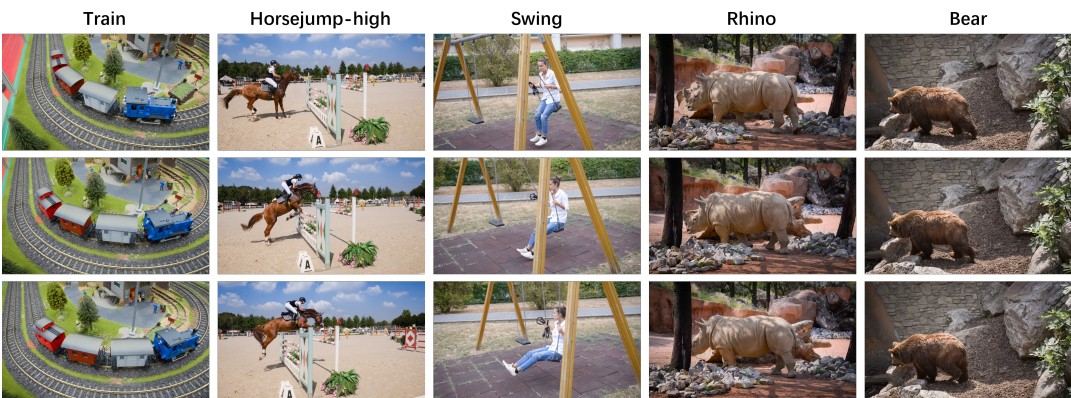

Figure 5: **Novel view synthesis from in-the-wild dynamic monocular videos.** Our method uses COLMAP-free dynamic monocular videos as input and reconstructs camera poses of all frames and Gaussian representation of the dynamic scene.

| Models | ATE↓ | $\text{RPE}_{\text{trans}} \downarrow$ | $\text{RPE}_{\text{rot}} \downarrow$ |
|--------|------|------------|----------|
| DROID_SLAM (Teed & Deng, 2021) | 0.175 | 0.084 | 1.912 |
| COLMAP (Schonberger & Frahm, 2016) | 0.213 | 0.164 | 5.312 |
| Robust-CVD (Kopf et al., 2021) | 0.360 | 0.154 | 3.443 |
| NeRF- (Wang et al., 2021) | 0.433 | 0.220 | 3.088 |
| BARF (Lin et al., 2021) | 0.447 | 0.203 | 6.353 |
| RoDynRF (Liu et al., 2023) | **0.089** | 0.073 | 1.313 |
| Ours | 0.165 | **0.069** | **1.028** |

Table 3: **Camera poses estimation results on the MPI Sintel dataset.** For the first 3 baselines, the dynamic parts are blocked out since they cannot handle dynamic scenes. Each baseline method is trained or run with its public code under the original settings and evaluated with the same evaluation protocol. The best results are highlighted in bold.

scene reconstruction task and the camera pose estimation task improves. Also, from Tab 5, we can find that the training time with relative pose initialization is the lowest among all other methods. These results reveal the effectiveness and efficiency of relative pose initialization on camera pose optimization, which helps the model to incorporate different frames into a compressed expression using the continuity between neighboring frames.

Second, we verify the effectiveness of our Triplane deformation field. From the 4th row and 5th row in Tab 4, it's obvious that our model's PSNR increases when imposing Triplane deformation to replace simple MLP deformation, which yields the necessity of incorporating our Triplane deformable field, especially when facing complex dynamic scenes.

Last, we test the advantage of our static Gaussian field, which replaces the original Gaussian representation in our model. It can be seen from the 3rd row and 5th row of Tab 4 that the triplane Gaussian field increases reconstruction quality and reduces pose error. This can be attributed to the weaker ability of our triplane Gaussian field, which reduces the possibility of geometric overfitting and thus ill-posed camera poses.

## 5 CONCLUSION

In this work, we propose Mono-DyGS, a novel end-to-end framework that jointly optimizes camera poses and dynamic scene representation on monocular videos. We demonstrate that previous works either deal with static backgrounds and dynamic foregrounds separately or require an extremely long training duration. We impose the relative pose initialization which significantly reduces the training time and improves the performance of camera pose estimation. Leveraging from the explicit representation of 3DGS (Kerbl et al., 2023), we propose a concise representation for both static backgrounds and dynamic foregrounds based on Hexplane. We show the effectiveness and robustness of our approach on challenging scenes like the DyCheck dataset. Thanks to the advantages of Gaussian splatting, our approach achieves rapid training and inference speeds.

| Pose | Deformation | Static | Novel View Synthesis | | | Pose Estimation | | |
|---|---|---|---|---|---|---|---|---|
| Initialization | Representation | Representation | mPSNR | mLPIPS | mSSIM | ATE↓ | RPE$_{trans}$↓ | RPE$_{rot}$↓ |
| | MLP | 3DGS | 21.09 | 0.562 | 0.331 | 0.501 | 0.312 | 10.329 |
| ✓ | MLP | 3DGS | 24.01 | 0.698 | 0.152 | 0.201 | 0.117 | 3.145 |
| ✓ | Triplane | 3DGS | 23.31 | 0.601 | 0.270 | 0.213 | 0.146 | 4.018 |
| ✓ | MLP | Triplane+3DGS | 24.36 | 0.702 | 0.137 | 0.198 | 0.102 | 1.543 |
| ✓ | Triplane | Triplane+3DGS | **24.52** | **0.755** | **0.101** | **0.165** | **0.069** | **1.028** |

Table 4: **Ablation results of different components on the iPhone Dycheck dataset and the MPI Sintel dataset.** The results are the averages over all scenes. The best results are highlighted in bold.

| Models | Times |
|---|---|
| DGSMarlbes | 4∼5h |
| RoDynRF | ≥ 20h |
| Shape-of-motion | 6∼7h |
| Ours w.o. pose initialization | ≥8h |
| Ours | **2∼3h** |

Table 5: **Training time on Nvidia Dycheck dataset.** The results are the averages over all scenes. The best results are highlighted in bold.

**Limitations**  Our relative pose initialization estimates camera poses sequentially, thereby restricting its application primarily to video streams or ordered image collections. Exploring extensions of our work to accommodate unordered image collections is promising for future research.

**Reproducibility Statement**  All experiments in this paper are reproducible. We are committed to releasing the source codes once accepted. Our code is built upon the Pytorch (Paszke et al., 2019).

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

## A  GEOMETRIC CONSTRAINTS ON DEFORMABLE FIELD

Given the 2D tracking trajectories of points $\{1, 2, \cdots, N\}$ provided by Co-tracker(Karaev et al., 2023), we use the ARAP principle to compute the corresponding deformed rotation by minimizing an ARAP loss:

$$R = \arg\min_R \sum_{i=1}^{T} \sum_{n=1}^{N} \sum_{m \in kNN(n)} \|R_{i,n}^{-1} P_{im} - R_{0,m}^{-1} P_{0n}\|, \tag{16}$$

where $R_{i,n}$ stands for deformed rotation at point $n$ at frame $i$; $P_{in}$ stands for the 3D coordinate of point $n$ at frame $i$ inferred from the given 2D tracking trajectories and depth; $kNN(n)$ indicates the top-k nearest points to point $n$. We further impose a rotation smoothness loss to supervise the rotation deformation of our deformable field:

$$\mathcal{L}_{\text{rot}} = \sum_{i=1}^{T} \sum_{n=1}^{N} \|R_{i,n} - \text{rotDeform}\left(W_0^{-1} \pi^{-1}\left(T_{0n}, d_{0n}\right)\right)\|, \tag{17}$$

where $T_{in}$ is the pixel coordinate of point $n$ at frame $i$; $\pi_{\mathbf{K}}$ represents the projection function from 3D space onto the pixel plane using camera intrinsics $K$; $W_i$ is the camera view transformation matrix at frame $i$, and rotDeform stands for the rotation deformation in our deformable Gaussian field.

Moreover, we introduce a spatial smoothness loss to enforce geometric consistency of the position deformation of our deformable field:

$$\mathcal{L}_{\text{center}} = \sum_{i=1}^{T} \| W_i^{-1} \pi^{-1} (T_i, d_i) - \text{posiDeform} \left( W_0^{-1} \pi^{-1} (T_0, d_0) \right) \|, \tag{18}$$

where posiDeform is the deformation of Gaussian's positions. In summary, the final loss for the dynamic part is composed of 3 components:

$$\mathcal{L}^d = \lambda_{\text{pho}} \mathcal{L}_{\text{pho}}^d + \lambda_{\text{rot}} \mathcal{L}_{\text{rot}} + \lambda_{\text{center}} \mathcal{L}_{\text{center}} \tag{19}$$

During optimization, the dynamic loss only passes gradients back to the Hexplane-based Gaussian field, while ignoring the camera poses.

