# OpenReview forum: "Learning Dynamic 3D Gaussians from Monocular Videos without Camera Poses"
_ICLR.cc/2025/Conference — ICLR 2025 Conference Withdrawn Submission_

### Official Review · Reviewer_daAw · 2024-11-01

**Soundness:** 2
**Presentation:** 1
**Contribution:** 2
**Rating:** 5
**Confidence:** 4

**Summary:**

This paper addresses the task of dynamic scene reconstruction, specifically developing a deformable 3DGS representation from an unposed monocular video. The proposed method first initializes camera poses by optimizing relative poses between adjacent frames via local 3DGS. To learn the global deformable 3DGS, a Hexplane-based encoder is employed to model both the static and dynamic regions in a unified manner. The authors evaluate the proposed method on diverse datasets and demonstrate its effectiveness and robustness.

**Strengths:**

- The method explores 3DGS for dynamic scene reconstruction with no pose prior.
- Comprehensive experiments on diverse datasets demonstrate its effectiveness in dynamic novel-view synthesis as well as camera pose estimation.

**Weaknesses:**

- Lack of novelty. The proposed method is more like a mixed bag which combines [1], [2], [3] and [4]. Specifically, the hexplane-based deformable Gaussian field has been explored by [1]. The relative pose initialization is adopted from [2]. The reprojection loss in Eq. 12 is similar to that of [3], while the depth alignment loss in Eq. 13 is similar to the Eq. 15 in [4]. The authors may have to make their contributions more explicit. Please also refer to Q1-2 in the question section.
- The authors may have to improve the clarity of writing. The paper is somewhat hard to follow due to factors such as missing necessary details (see Q3-6), inconsistencies (see Q7), and missing citations (Q8).
- The limited qualitative results raise another concern. It would be preferable to include videos or real-world demonstrations as additional supplementary material. Also see Q9.


[1] Guanjun Wu, et al. "4D Gaussian Splatting for Real-Time Dynamic Scene Rendering." Proceedings of the IEEE/CVF Conference on Computer Vision and Pattern Recognition (CVPR). 2024.

[2] Yang Fu, et al. "COLMAP-Free 3D Gaussian Splatting." Proceedings of the IEEE/CVF Conference on Computer Vision and Pattern Recognition (CVPR). 2024.

[3] Yu-Lun Liu, et al. "Robust Dynamic Radiance Fields." Proceedings of the IEEE/CVF Conference on Computer Vision and Pattern Recognition (CVPR). 2023.

[4] Jiahui Lei, et al. "MoSca: Dynamic Gaussian Fusion from Casual Videos via 4D Motion Scaffolds." arXiv preprint arXiv:2405.17421. 2024.

**Questions:**

1. The static Gaussian field and the deformable Gaussian field are optimized separately using mutually exclusive masks, but why do the authors claim that the proposed hexplane-based Gaussian field is a _unified_ representation? From my perspective, 4DGS appears to provide a more ‘unified’ representation.
2. Is the deformable Gaussian field optimized using only foreground dynamic masks? In other words, only dynamic regions of the deformable Gaussian field are supervised? Then how to ensure that Gaussians in static regions would not be affected by the deformable field when rendering from a novel view?
3. How are 3D Gaussians initialized before optimizing the Hexplane-based Gaussian field?
4. Please provide more details on how to minimize the objective defined in Eq. 16.
5. How are the static regions obtained in each frame?
6. It would be beneficial to provide a mathematical form of the _rotDeform_ in Eq. 17.
7. How many datasets are used for evaluation? The authors list three: DyCheck, NVIDIA DynamicNeRF, and MPI Sintel (Line 86-87), but an additional DAVIS dataset is also mentioned (Line 413).
8. Including relevant citations for the scale-invariant loss and ARAP loss would be helpful.
9. I am concerned about the qualitative results in Fig. 4, especially the last two columns, which empirically do not align with the quantitative results in Tab. 2. There is an urgent need for additional qualitative comparisons presented in the form of videos.

- Please check the format of citations:
> When the authors or the publication are included in the sentence, the citation should not be in parenthesis using \citet{}.
- Please check the format of notations. For example, it is recommended to format the high-dimensional spatial-temporal feature _in bold_ as $\mathbf{f}_d$.
- A non exhaustive list of typos:
	- Line 69: vides -> videos
	- Line 137-138: files -> fields

---

### Official Review · Reviewer_MUrG · 2024-11-03

**Soundness:** 2
**Presentation:** 2
**Contribution:** 2
**Rating:** 3
**Confidence:** 4

**Summary:**

The paper is motivated to model dynamic Gaussian Splatting scenes without knowing camera poses. The authors point out that the previous approaches typically separately model the static and dynamic regions, leading to prolonged training time and potentially suboptimal reconstruction. In response to these problems, the authors propose to initialize the camera poses with pair-wise relative camera poses, and a unified representation using Hexplane for modeling the static and dynamic regions together. Depth and optical flow priors are also introduced to regularize the motion further.

**Strengths:**

The problem of focus is clearly motivated. Reconstructing a dynamic scene from a monocular sequence is becoming more important and is a nontrivial task. The proposed pipeline is able to outperform previous methods for most tested scenes. The diagrams are clear and easy to understand. There are enough visuals to illustrate the improved render quality. Thorough baseline comparisons are included to demonstrate the performance improvements.

**Weaknesses:**

Although the proposed approach outperforms existing approaches, it seems that the approach is more like an engineered combination of previous approaches. The initialization of the pairwise camera pose is identical to that of the camera initialization in DGMarbles. The Hexplane representation of a Gaussian scene seems to be inherited from the 4DGS method. Currently, it is hard to identify the main difference of each component in the proposed approach from the previous methods. The paper can be further improved if the merits are highlighted and with detailed discussions to distinguish itself from baselines.

Besides, the paper's claim on not "disentangling static and dynamic regions using two separate representations" seems questionable. Although the static and dynamic parts do share the same Hexplane representation, they are still supervised differently, leading to different treatments for the two parts. The approach used a complex combination of loss terms, which are also not ablated in the experiments, so it is hard to know whether these loss terms are necessary or if they could potentially be part of the byproduct of using a Hexplane representation.

Overall the novelty of the approach seems limited without further clarifications. The paper can be potentially further improved by addressing this problem and including more ablation studies on the loss term to promote understanding of the system.

**Questions:**

It would be helpful if the authors could address the following questions:
1. What is the main difference between the proposed pairwise camera pose estimation compared to the camera marbles initialization in DGMarbles? It would be helpful to clarify or discuss this in the paper.
2. The Hexplane representation seems to be one of the main contributions of the paper. If that is the case, it would be helpful to discuss its relation to 4DGS in Section 3.3 and highlight their differences.
3. Similar to DGMarbles, the proposed approach supervises the motion field with a tracked 2D trajectory from CoTracker. Why is this supervision chosen in opposed to the optical flow that was used in the static regions? Would it bring any difference if both regions were supervised with 2D trajectory or optical flow?
4. The ablation study only studies the differences in representation. However, the proposed method has a heavy emphasis on loss regularizations, which likely heavily contributed to the improved performance. How would the system perform without each regularization term? It would be helpful to include these results to bring intuitions on how and why the chosen Hexplane representation brings merit.
5. The paper reads to be a carefully engineered system comprised of components from previous approaches. It is difficult to locate the merit or intuitions from the paper. The paper can be further improved if the novelties are highlighted and discussed more in detail.

---

### Official Review · Reviewer_uw8v · 2024-11-03

**Soundness:** 3
**Presentation:** 3
**Contribution:** 3
**Rating:** 5
**Confidence:** 5

**Summary:**

This work presents a framework to efficiently reconstruct dynamic scenes from casually captured monocular videos. Similar to many concurrent works, the main method also uses Gaussian splatting as a 3D representation. In this framework, a camera estimation module is introduced to obtain frame-wise camera poses. Deformations of Gaussians are represented using a HEX-Plane representation. Extensive experiments are conducted on datasets including Dychec, NVIDIA, and Sintel.

**Strengths:**

1. Promising experimental results demonstrate the effectiveness of this work.
2. Camera pose estimation with relative initialization and joint optimization is novel.
3. Splitting dynamic and static objects in scenes for optimization can reduce artifacts in reconstruction.

**Weaknesses:**

1. Limited technical novelty. Combining HexPlane representation with Gaussian Splatting does not seem novel, as many published works have combined TriPlane representation with Gaussian Splatting.

2. Lack of justification for using HexPlane. Although disentangling dynamic and static objects in scenes is sound, the adoption of HexPlane representation is not well-presented. A simple Fourier series, as used in Splatter-a-Video, can also represent Gaussian dynamics in the center position and rotations. While HexPlane introduces significantly more computation and storage overhead, it is vital to justify the use of HexPlane over a simple Fourier series.

3. Although this work is concurrent with similar Gaussian video representations, such as Splatter-a-Video [1] and GFlow [2], discussion on these works is still necessary, as these two works have been publicly available on arXiv for about four months before the ICLR submission deadline.

4. The relative camera pose module is designed only with depth priors, focusing on relative camera movement between two frames. This setting is consistent with that in Dust3R; why not directly apply Dust3R?

5. Evaluation metrics are too limited. As the relative camera poses are initialized and further jointly optimized in this work, why not quantitatively evaluate the camera pose accuracy on the Sintel dataset?

6. Typos exist, e.g.  line 080 and line 138.




[1] "Splatter a Video: Video Gaussian Representation for Versatile Processing.",  NeurIPS 2024

[2] "GFlow: Recovering 4D World from Monocular Video", arxiv 2024

**Questions:**

please refer to the weakness part.

---

### Official Review · Reviewer_rJ5o · 2024-11-04

**Soundness:** 4
**Presentation:** 4
**Contribution:** 3
**Rating:** 8
**Confidence:** 4

**Summary:**

The paper proposes a new method for dynamic scene reconstruction. The main novelty is in using the hexplane representation without known camera poses. Results show superior performance.

**Strengths:**

- The idea is interesting and important. The topic itself has gained a lot of attention in recent years.
- The novel idea of hexplane representation together with camera pose initialization with additional optimization is appreciated.
- Also, using priors such as depth and optical flow is meaningful.
- The method outperforms other methods on this task, sometimes even methods that assume camera poses.
- The paper is well-written and easy to follow.

**Weaknesses:**

- The method would have been stronger if it didn't have to assume the given camera intrinsics.
- More qualitative results would make the paper better.

**Questions:**

It would be interesting to add even more ablation studies to understand which parts of the method are more important.

---

### Note · Authors · 2024-11-15

I have read and agree with the venue's withdrawal policy on behalf of myself and my co-authors.